# Using Sensor Graphs for Monitoring the Effect on the Performance of the OTAGO Exercise Program in Older Adults

**DOI:** 10.3390/s22020493

**Published:** 2022-01-10

**Authors:** Björn Friedrich, Carolin Lübbe, Enno-Edzard Steen, Jürgen Martin Bauer, Andreas Hein

**Affiliations:** 1Assistance Systems and Medical Device Technology, Carl von Ossietzky University, Ammerländer Heerstraße 114-118, 26129 Oldenburg, Germany; carolin.luebbe@uni-oldenburg.de (C.L.); enno-edzard.steen@uni-oldenburg.de (E.-E.S.); Andreas.Hein@uni-Oldenburg.de (A.H.); 2Center for Geriatric Medicine, Agaplesion Bethanien Hospital, University of Heidelberg, Rohrbacher Straße 149, 69126 Heidelberg, Germany; bauer@nar.uni-heidelberg.de

**Keywords:** motion sensors, sensor network, pervasive computing, ambient intelligence, frail older adults, exercise training, physical performance, healthcare

## Abstract

The OTAGO exercise program is effective in decreasing the risk for falls of older adults. This research investigated if there is an indication that the OTAGO exercise program has a positive effect on the capacity and as well as on the performance in mobility. We used the data of the 10-months observational OTAGO pilot study with 15 (m = 1, f = 14) (pre-)frail participants aged 84.60 y (SD: 5.57 y). Motion sensors were installed in the flats of the participants and used to monitor their activity as a surrogate variable for performance. We derived a weighted directed multigraph from the physical sensor network, subtracted the weights of one day from a baseline, and used the difference in percent to quantify the change in performance. Least squares was used to compute the overall progress of the intervention (n = 9) and the control group (n = 6). In accordance with previous studies, we found indication for a positive effect of the OTAGO program on the capacity in both groups. Moreover, we found indication that the OTAGO program reduces the decline in performance of older adults in daily living. However, it is too early to conclude causalities from our findings because the data was collected during a pilot study.

## 1. Introduction

The frailty syndrome describes the condition in which the muscular performance and wellbeing of geriatric people continuously decline. This can result in a loss of independence and an increased risk of falls. Particularly, in the group of older adults, falls can have serious physical consequences and weaken the confidence in one’s abilities, leading to a loss in independence. The exercise program named OTAGO aims to reduce the risk of falling with the help of an exercise program comprising 17 exercises tailored for the individual participant [1,2]. It includes exercises in increasing difficulty to strengthen the lower-limb muscles, exercises to improve balance, and a training plan for regular walks. Risk assessment and performance monitoring can be conducted using well-established geriatric mobility assessments, which are performed under the supervision of healthcare professionals. Two common and validated mobility assessments are the Timed Up and Go (TUG) and the short physical performance battery (SPPB), each measuring the dimensions gait speed, lower-limb strength, and balance [3,4].

Supervised assessments are often time-consuming and create a test situation in which the participants attempt to do their best (Hawthorne effect) [5]. However, this may lead to results that reflect capacity rather than the participant’s performance [6]. The gap between capacity and performance of the mobility of older adults is a well-known fact in geriatrics research and studies show that performance is a better indicator of health condition than the capacity. The example gait speed shows that the self-selected gait speed is a good predictor for mortality rates and daily ambulatory activity in older adults [7,8,9]. Our objective is to measure the performance of older adults in daily life and compare the performance of the intervention group doing the OTAGO exercise program and the control group not doing the exercise program. We used passive infrared (PIR) motion sensors installed in the domestic environments and a weighted directed multisensor graph as an activity measure to achieve our objective. We used activity as surrogate variable for performance. PIR motion sensors were chosen because they are more privacy-preserving than, e.g., cameras. The capacity was measured by monthly standardized geriatric assessments SPPB and TUG. Our data was collected during the pilot observational “OTAGO” study with 20 (pre-)frail participants (aged: 84.75 y, SD: 5.19 y) over 10 months.

## 2. State of the Art

In this section, we give a brief overview of recent research. At first, we introduce the state of the art in geriatric medicine for assessing the mobility and risk for falls in older adults. We use these assessments as capacity measure in testing situations. Afterwards, we mention the computer science approaches to estimate mobility, gait parameters, and the falling risk.

The TUG test and the SPPB are two common validated mobility assessments in geriatric medicine [3,4]. Both assessments assess the three dimensions gait speed, lower-limb strength, and balance. The procedure for the TUG test is as follows: the subject sits on a standardized chair, on the command “go”, the subject stands up, walks three meters, then turns around, walks back to the chair, and sits down again. The supervisor measures the time in s using a stopwatch. The measurement starts when the supervisor gives the command and ends when the subject’s buttock touches the chair again. Depending on the time, the subject’s mobility and risk of falling can be categorized as

≤10 s: normal, no mobility impairment.11–19 s: minor mobility impairment, not relevant in everyday life.20–29 s: mobility impairment.>30 s: severe mobility impairment, need for intervention.

If an assistive device for walking is used, the usage must be recorded and the assistive device must be used in any retest. The dimension gait speed is assessed by the three-meter walk, the dimension balance by standing up from the chair and turning around, and lower-limb strength by standing up from the chair and sitting down again. While the TUG assesses the three dimensions implicitly, the SPPB has separate items for each dimension which are scored independently. The SPPB is comprised of the three items:Stance, semi-tandem stance, full-tandem stance (balance).Four-meter walk (gait speed).Five-time chair rise (lower-limb strength).

Each item can be scored from 0 points to 4 points, and a maximum score of 12 points can be achieved. The time needed for the four-meter walk and the five-time chair rise is measured in s by the supervisor using a stopwatch, and the balance is scored by the time the subject can stand in the different stances. The cut-off value for fit and frail people is ≤9 points and the minimal clinically important difference is 0.5 points [10,11]. For medical purposes, the MCID is more meaningful than a statistical significance. Both assessments assess a limited number of parameters, e.g., jumping performance, walking backwards, lying down, and rolling are neither included in the SPPB nor in the TUG. Naturally, assessment results measure the form on the day. Moreover, it has been shown that energy and fatigue from, e.g., answering questionnaires could have an negative impact on the assessment results [12].

Researchers utilized various sensors to estimate gait parameters, mobility, and the risk of falling in older adults. There are mainly two different approaches. The first approach is to support the assessment using sensors and computer science technologies, and the second approach is to measure the parameters in daily life. The measurements in daily life are the closest to measure the performance instead of the capacity.

The movement during the TUG test can be found in several situations in daily life, e.g., standing up from a chair, sitting down on a chair, and walking a short distance are everyday activities. Therefore, inertial measurement units (IMU) were used to detect the TUG movements in everyday life and to estimate the time of the movements [13]. Moreover, an approach to estimate the scores of mobility assessment based on real-life IMU data has been made in [14]. The systems developed in [15,16] are supposed to enable people to perform assessments in an unsupervised fashion.

As gait parameters and body movements can be measured directly when a sensor is attached to the body, it is not far to seek to utilize IMUs. Most of the research using wearable sensors are focused on gait parameters [17,18,19]. Studies using IMUs showed that the results are comparable to the gold-standard measurements [20,21,22]. In combination with deep neural networks, IMUs show promising fall detection and fall risk prediction results [23,24,25,26]. Even for older adults suffering from neurological disorders and multiple sclerosis, the results are convincing [27,28]. The approaches described in [29,30,31] were able to measure gait parameters and walk characteristics using IMU and force sensor data. IMUs ambient sensors installed in domestic environments are in the focus of researchers. We only mention approaches that use privacy-conserving sensors because the sensors are installed in private areas. Privacy is of great concern, and cameras or light detection and ranging (LiDAR) sensors are too intrusive. Moreover, unobtrusive ambient sensors are well-accepted among older adults [32,33]. Sensor networks installed in smart homes were used to estimate the self-selected gait speed in domestic environments [34]. Moreover, utilizing homogeneous networks of PIR motion sensors for the same purpose show precise estimation as well [35,36,37,38]. Another approach using radio signals estimated not only the gait speed but also the stride length as gait parameter [39].

The mentioned approaches primarily focus on gait parameters, and the technology-assisted mobility assessments place the participants in test situations again. However, our approach is more versatile and does not depend on special movements.

## 3. Materials and Methods

### 3.1. Study Design

The primary aim of the study was to analyze the “effect of the fall prevention intervention program OTAGO on older persons with frailty and prefrailty” and compare the conventional assessment of mobility and independence of older adults in (pre-)frail condition with sensor-based documentation. The study was conducted in 2014 and 2015 for a period of 10 months. A total number of 20 participants (17 female, 3 male), aged 84.75 y (SD: 5.19 y), were recruited at local nursing homes and among participants of previous studies. They were randomly assigned to the control and the intervention group. The cohort size was limited by human resources and funding. The inclusion criteria were as follows:Age ≥ 75.At least pre-frail by the definition of Fried (Frailty Index ≥ 2) [40].Living alone within the city limits of Oldenburg.Able to move inside the flat as needed.

The exclusion criteria were as follows:Not able to move inside the flat as needed.Keeping pets that move freely inside the flat.Living with other people.Severe visual impairment.Contraindication against the OTAGO exercise program.Unable to understand the purpose of the study and the study itself.

The average participation time was 36.5 weeks (no follow-up). Two participants passed away during the study. The participants in the intervention group were required to independently perform the 30 min OTAGO exercise program once a week and to walk for at least 30 min every day [1]. The exercise program includes five exercises to strengthen the lower-limb muscles, twelve exercises to improve balance, and a training plan for regular walks. The exercises could be performed in up to four levels of difficulty and were individually tailored to the participant. The execution and tailoring were controlled and carried out by licensed physiotherapists. In addition, in-home assessments were conducted once a month, during which the exercise selection and intensity were adjusted if necessary. The average time between two in-home assessments was 31.3 d (SD: 5.3 d). The in-home assessments included, but were not limited to, the mobility assessments Tinetti [41], SPPB, TUG, and the standardized questionnaires EQ–5D–5L [42], Mini Nutritional Assessment [43], and Instrumental Activities of Daily Living [44], amongst others. The questionnaires and assessments were selected to obtain a holistic view of the participants and their condition. The baseline was established by the first assessments at the beginning of the study; 10.35 assessments are available on average per participant. Table 1 and Table 2 show a selection of characteristics of the study cohort and the used sub-cohort at baseline and the end of the study.

### 3.2. Data Acquisition

The sensor-based documentation was realized by installing an ambient multi-component sensor system in the domestic environments of the participants. In addition, two wearable sensors were given to them as well. The sensors for the ambient multi-component sensor network were chosen with privacy and acceptance among older adults in mind [32,33]. The chosen sensors were PIR motion sensors, door contact sensors, a four-key button, a vibration sensor, and power consumption sensors. The PIR motion sensors had a cool time of 8 s when motion could not be detected. The PIR motion sensors were installed so that they could cover the rooms of the flat and a person entering a room can be detected as soon as possible. Additional ones were placed so that a specific path in a room or between two rooms is covered, e.g., the way from the bedroom to the kitchen. One PIR sensor was placed in the bathroom right over the lavatory flush to detect if the resident used the toilet. Moreover, one redundant sensor was installed in every flat. The PIR motion sensors had a cool-down time of 8 s and measured motion in 8 s intervals. The door contact sensors were installed at the entrance doors to the flats and the fridge to detect when a door was opened. The button was placed at the front door and the participants were asked to push the button if somebody else was entering or leaving the flat. The door contact sensor and the button were combined to detect if another person was in the flat. The vibration sensor was placed between the slatted frame and the mattress of the resident’s bed. Using the vibration sensor, sleep times information can be measured more accurately. The power-consumption sensors were attached to several appliances in the flat; these sensors were attached to the socket of the appliance and continuously measured the power consumption. All sensors transmitted their data by a wireless connection to a base station. The two wearable sensors were a Columbus V990 GPS sensor and an IMU of type Shimmer3r [45,46]. The participants were asked to take the GPS sensor with them when leaving the flat. The IMU was given to the participants on every assessment day and collected one week after, and the participants were responsible for transferring the data and charging the IMU by placing it on a docking station. Only the data from the sensor network were used for this research, and no data from the wearable sensors. The sensor network differs from flat to flat because of the different topologies.

### 3.3. Sensor Graph

We modeled the flats as weighted directed multigraphs. Each vertex represents one motion sensor or one room. We used directed graphs because we considered the order of the sensor events, e.g., a transition from the living room to the kitchen is different to a transition from the kitchen to the living room. In this regard, we needed to use multigraphs, because there must be one edge for each transition. The weights were computed according to the order and number of motion sensor events, e.g., if there was a motion sensor event of the kitchen sensor followed by an event from the living room sensor, 1 was added to the weight of the edge representing the transition from the kitchen to the living room. Hence, the primary diagonal of the adjacency matrix represents motion in one room. The motion in the same room was a loop in the sensor graph. The transitions are likely to be motion, and motion is essential for analyzing the effect of the OTAGO program. The primary diagonal had the largest values, because the participants were in one room and they were not changing rooms most of the time. To increase the importance of transitions, we used a weight matrix W with weight 1 on the primary diagonal and 1.5 elsewhere. In mathematics, a graph is defined as an ordered triple of vertices, edges, and an incident function $G=\left(V,E,\phi \right)$, where $V=\{s_{0},\ldots ,s_{n-1}\}$ is the set of vertices or sensors and *n* is the number of motion sensors in the flat, *E* is the set of edges, and $\phi :E\subseteq \{\left(s_{i},s_{j}\right)|\left(s_{i},s_{j}\right)\in V\times V,0\leq i,j<n\}$ is the incident function. Please note that we allowed loops in the graph by the definition of ϕ. We also introduced the edge weight update function $f_{w}\left(G,\;i,\;j\right)\;=G+T$, where $G\in R^{n\times n},T\in {\left\{0,1\right\}}^{n\times n}$ and 0 everywhere except for $t_{i,j}=1$. The following Table 3 and Table 4 show the baseline and a random day adjacency matrix of participant 6.

The sensor graph for the flat in Figure 1 is shown in Figure 2. The edge weights were the baseline weights. Each vertex is connected to all other vertices with two edges in the sensor graph. The example shows that the most activity was found in the living room and the highest number of transitions was from the living room to the hallway. The graph also contains transitions that were impossible due to the topology of the flat, e.g., in the example flat, a direct transition from the bedroom to the bathroom was not possible. The impossible transitions have the weight 0.

### 3.4. Difference Function

Before computing the difference, both matrices, the baseline adjacency matrix *B*, and the daily adjacency matrix *D*, were elementwise multiplied with the weight matrix *W*. The difference is the standard subtraction defined over the ring of matrices:(1)$$\mathrm{diff}\left(B,D\right)=\sum_{i,j=0}^{n-1}b_{i,j}-d_{i,j}$$

Finally, we converted the difference to percent $p_{d}$ of the baseline:(2)$$p_{d}=\frac{\mathrm{diff}\left(B,D\right)}{\sum_{i,j=0}^{n-1}b_{i,j}}$$

### 3.5. Preprocessing

Before the algorithm was applied, the data must be preprocessed and filtered on participant scale and day scale. The participants were filtered by the following three exclusion criteria:Passed away during the study.Change in living situation.Incidents compromising mobility.

Two participants were excluded according to criterion 1, they were deceased 9 months after study began. According to criterion 2, two participants were excluded because the first participant moved to a nursing home after 4.5 months and the second participant had the flat renovated after 7.5 months. According to criterion 3, one participant was excluded because the participant had a severe fall incident in the first month of the study and had to use a wheelchair after rehabilitation. Eventually, our analysis contained 15 participants; nine were from the intervention group and six from the control group. The days where the participant was absent were filtered, and 115 d were filtered, then overall, 4534 d (per participant: mean 302.27 d, SD 20.74 d) were left for our analysis. The reasons for being absent were vacation, short-term care, and hospitalization. One participant went on vacation and relatives were living in the flat during that time.

### 3.6. Statistical and Computational Methods

Before the analysis, we had to validate that our sub-cohort had no bias after applying the exclusion criteria. The dependent variables were the tests and questionnaire results. Thus, the dependent variables were all on metric scale and so the assumption for the test holds. We also report if there is a bias in the control group. We used the Wilcoxon signed-rank test to report if the difference in activity and the improvement in the mobility assessment scores were statistically significant. We carefully checked all assumptions for applying the test. Our computational method for capturing the tendency of the performance was the method of least squares. We used it to compute the best parameters a,b of a linear function.

## 4. Results

The statistical results showed that the disposition of the cohort and the sub-cohort were the same to a significance level of α=0.01 (Appendix A Table A1). The disposition of the intervention and control group were the same to a significance level of α=0.01 (Appendix A Table A2). The improvements in the SPPB and TUG score were not statistically significant for the control group and neither was there a difference in the intervention group.

The results for the geriatric assessments are shown in Figure 3 and Figure 4. The fitted lines show that the capacity of both cohorts was increasing over time. The capacity of the intervention group was increasing faster than the capacity of the control group. The SPPB scores of the intervention group improved by 0.16 points, and the control group by 0.06 points. The lines of the relative TUG times were decreasing because the shorter the time, the better the mobility of the participant. The TUG time of the intervention group improved by −0.10 s, and the control group by −0.08 s. The differences were negative but meant an improvement in the test, and so in capacity. In Figure 5, the fitted lines of the activity difference to the baseline in % are shown. Both lines had a negative slope; the slope of the control group was smaller. The lines show that the activity in the flat of the control group was decreasing faster than in the flats of the intervention group. The difference between the intervention group was changing by −0.02% and of the control group by −29.35%. Both differences were negative, and that meant less activity in the flat. The parameters show that the slope of the fitted line of the control group is smaller by a factor of 10. The difference ($H_{0}$: the fitted activity of the control group has the same median as the fitted activity of the intervention group) is statistically significant to a significance level of α=0.01. The difference of the study duration (304 d) was analyzed and the test statistics of the Wilcoxon signed-rank test were −4.76<W=46,360, $p<6.74\times 10^{-52}$.

## 5. Discussion

The analysis of the geriatric mobility assessments had expected results. The intervention group performing the OTAGO exercise program showed a greater improvement in capacity than the control group, and the analysis of the differences showed interesting results. The improvement in the control group may be caused by a learning effect. The differences of the sensor graphs were nearly stable for the intervention group, whereas the differences were decreasing for the control group. The analysis of the assessments showed that the magnitude of the difference of both groups is of factor 2, but the differences of the sensor graphs are a factor of 10. Considering the difference between capacity and performance places a different complexion on the difference analysis results. The results indicate that the OTAGO exercise program improves the capacity of older adults but not the performance. Furthermore, an improvement in capacity seems to not affect the performance in real life. On the one hand, an improvement in capacity is not visible because the person’s actual performance is sufficient for daily life activities, e.g., a person would not run to the kitchen to get a cup, even though it would be faster. On the other hand, a decrease in capacity may not be visible until the level of capacity is decreasing below the level of performance. Our results support the findings that the capacity does not reflect the performance and the Hawthorne effect [5,6]. However, it is too early to conclude causalities from these results.

Several factors influence the activity and mobility in domestic environments. One factor may be the motivation to perform a specific task, e.g., going to answer the phone may be faster than going to the kitchen to return a used cup. During the study, some motion sensors were rearranged, disassembled, or dropped by the inhabitant of the flat. Moreover, the base station for recording the sensor data was disconnected or down. This led to faulty and corrupted data and complete loss of data. Hence, the groups are imbalanced after filtering. There are three fewer participants in the control group than in the intervention group. Another important limitation is the presence of other people in the flat. All participants were living alone in the flats, but they may have had visitors such as relatives, nurses, and workers. Filtering all the time where other people were present in the flat was not possible and the participants were not using the four-key button reliably. We tried to mitigate this limitation by fixing the weights of impossible transitions to 0, but if other people were in the same room, the activity may be overestimated. Another limitation is that the PIR sensors can only measure motion and not other aspects of mobility such as balance and lower-limb strength. The system could detect changes, and these changes could be used as an indicator for assessing the mobility using a standardized geriatrics assessment supervised by a licensed physiotherapist.

## 6. Conclusions

In this article, we introduced sensor graphs generated from PIR motion sensor networks in domestic environments that can quantify movement within, and transitions between, rooms and thus allow drawing inferences about the resident’s performance. Our results indicated that sensor graphs generated from a PIR motion sensor network can be used to monitor the performance progress of older adults. In addition, we found an indication that the performance of older adults, who completed the OTAGO exercise program, declines slower than the performance of older people who did not complete the exercises, even though the capacity of both groups increased. The results also suggest that the OTAGO exercise program may improve the capacity of older adults but not their performance. However, it is too early to conclude causality from these results; hence, this will require further clinical studies and a sophisticated strategy for filtering the times when more than one person is in the flat.

## Figures and Tables

**Figure 1 sensors-22-00493-f001:**
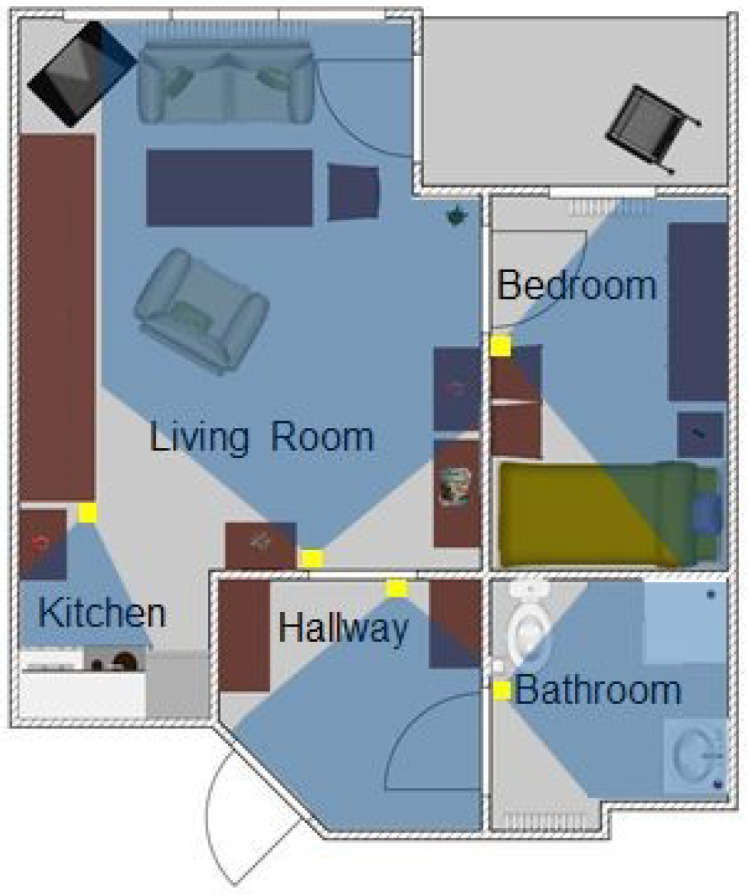
The flat of participant 6. The yellow boxes indicate the position of the motion sensors, and the blue shades the recorded area.

**Figure 2 sensors-22-00493-f002:**
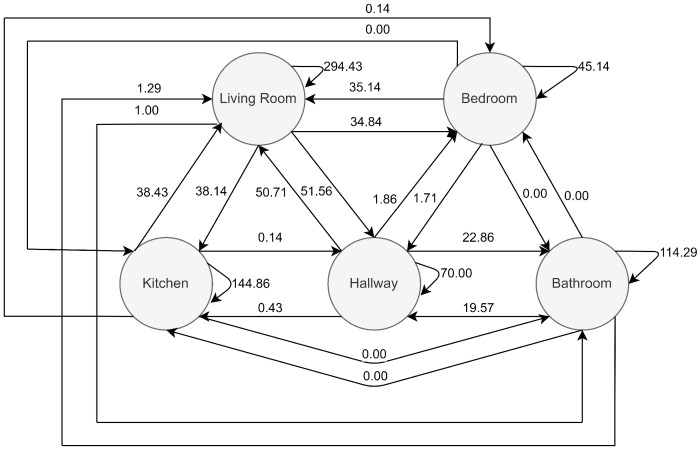
The sensor graph of the flat (Figure 1) of participant 6 with the baseline weights.

**Figure 3 sensors-22-00493-f003:**
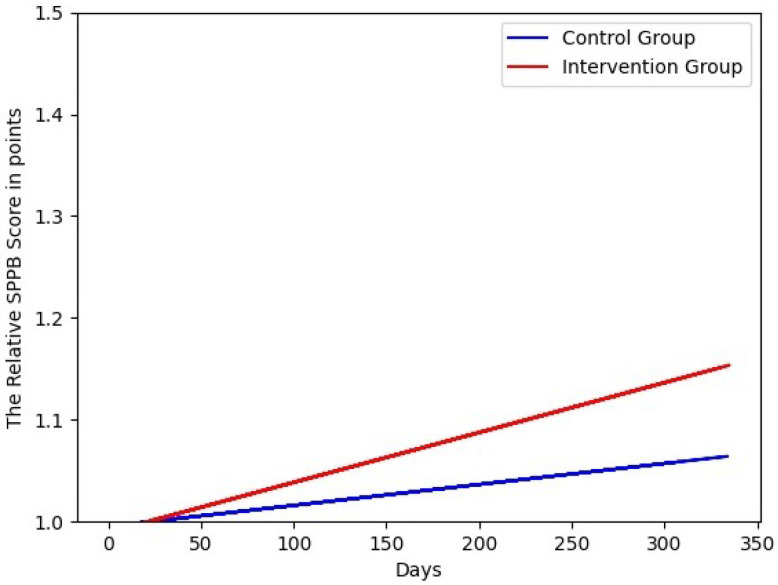
The fitted lines of the relative SPPB scores in points. The scale on the *y*-axis is the MCID.

**Figure 4 sensors-22-00493-f004:**
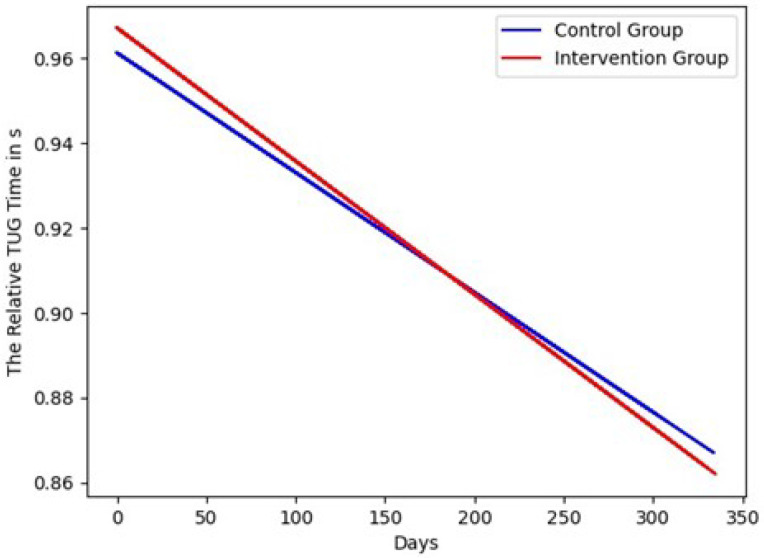
The fitted lines of the relative TUG time in s.

**Figure 5 sensors-22-00493-f005:**
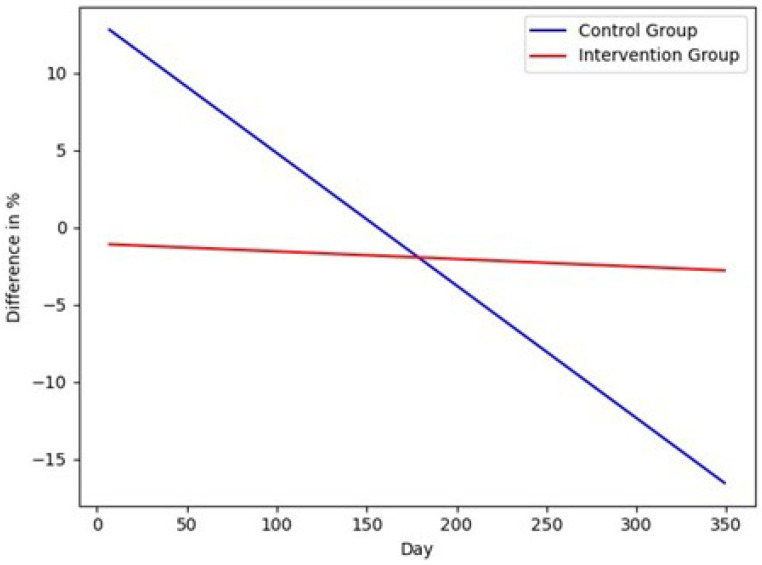
The fitted lines of the difference in % of the daily sensor graphs.

**Table 1 sensors-22-00493-t001:** The baseline/end characteristics of the cohort.

n = 20/18 (m = 3/3, f = 17/15)	Age (y)	Frailty Index (Points)	SPPB (Points)	TUG (s)
Mean	84.75/85.44	1.90/2.00	5.95/6.61	17.87/16.12
SD	5.19/4.92	0.72/0.97	2.33/2.85	5.33/5.85
Range (min–max)	76.00–92.00/ 77.00–93.00	1.00–3.00/ 0.00–4.00	3.00–11.00/ 2.00–12.00	11.16–31.63/ 8.15–30.06

**Table 2 sensors-22-00493-t002:** The baseline/end characteristics of the used sub-cohort at the beginning of the study.

n = 15/15 (m = 1/1, f = 14/14)	Age (y)	Frailty Index (Points)	SPPB (Points)	TUG (s)
Mean	84.60/85.20	1.80/1.73	6.53/7.07	16.81/14.87
SD	5.57/5.41	0.68/0.80	2.36/3.01	4.35/5.82
Range (min–max)	76.00–92.00/ 77.00–93.00	1.00–3.00/ 0.00–3.00	3.00–11.00/ 2.00–12.00	11.16–24.06/ 8.15–30.06

**Table 3 sensors-22-00493-t003:** The matrix is the baseline adjacency matrix of participant 6.

	Bedroom	Bathroom	Kitchen	Living Room	Hallway
**Bedroom**	45.14	0	0.14	34.84	1.86
**Bathroom**	0	114.29	0	1	22.86
**Kitchen**	0	0	144.86	38.14	0.43
**Living room**	35.14	1.29	38.43	294.43	50.71
**Hallway**	1.71	19.57	0.14	51.86	70

**Table 4 sensors-22-00493-t004:** The adjacency matrix of a random day of participant 6.

	Bedroom	Bathroom	Kitchen	Living Room	Hallway
**Bedroom**	45	0	0	34	1
**Bathroom**	0	98	0	1	25
**Kitchen**	0	0	196	39	1
**Living room**	32	1	39	230	68
**Hallway**	3	25	1	66	64

## Data Availability

The data are not publicly available due to privacy concerns.

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
