# Peer review of "Using Sensor Graphs for Monitoring the Effect on the Performance of the OTAGO Exercise Program in Older Adults"

_sensors, 2022, doi:10.3390/s22020493_

Round 1

Reviewer 1 Report

Comment 1
The article doesn't mention what OTAGO stands for. It would be preferable to specify the meaning for researcher that are not familiarized with it. Also, it would be good to specify why choosing OTAGO. Were there any indications that these type of exercises would have an impact?

Comment 2
Lines
239 Before the analysis we had to validate that our sub–cohort after applying the
240 exclusion criteria have not had a bias.
are not needed, because it's almost identical with and are too close to eachother:
232 Before the analysis, we had to validate that our sub–cohort had no bias after
233 applying the exclusion criteria.

Comment 3
As a general observation, some of the phrases are too long and difficult to understand. But overall, the clarity is good.

Reviewer 2 Report

GENERAL COMMENTS:

This paper use sensor graphs to monitor the effect on the performance of the OTAGO exercise program. The idea of using sensors to monitor mobility performance automatically is novel. The experiments were performed in real-world flats, including participants of advanced age, which is suitable for demonstrating the effects. However, there are also some significant issues. The introduction does not provide enough information. OTAGO should be well explained with several sentences instead of simply citing a paper. In the method section, the advantage of using sensor graph seems not obvious because figure 2 is more complex than the table following line 202 (missing table title). In the Discussion section, it will be better to compare with other studies, making the discussion more substantial.

SPECIFIC COMMENTS:

Lines 123-133: the bullets should be a different style.

Lines 174-177: Were the IMU and GPS data was used? If not, write some sentences around here. Otherwise, it makes the reader confused.

Line 227: 115d, d is not easy to be understood as “day”. The same issue in Line 77, s is not easy to be understood as “second”.

Line 87: “ [10] used Inertial Measurement” is not a comment style to cite a paper.

Line 249: Is it a statistically significant improvement by improving 0.02 s?

Reviewer 3 Report

Sensors: Sensor Graphs / Otago

This study presents a novel approach in evaluation a fall prevention / physical activity promotion using a sensor network installed in participants’ homes. The motion sensor system described here is very novel and has the ability to give insight into physical activity behavior within persons’ home environment. The study was conducted in 20 older persons, half of which received the OTAGO exercise programme as an intervention. Assessments were carried out on a monthly basis for about 300 days (on average).

Overall, this study shows merit and is based on a high effort. Unfortunately, I have many concerns on the organization of the manuscript itself, the results and their interpretation as well as rather many uncertainties. My recommendation is to reject the manuscript for those reasons. I will provide more explanation below.

1) the sample is unclear to me. In line 118-120 it is stated that participants were recruited in nursing homes. How can nursing home residents be defined as “living alone”? Several people enter rooms of nursing home residents, among these other residents, carers, other personnel. This would introduce serious problems to the motion sensors (as you state in the limitations, other persons cannot be ruled out in the results). A clearer understanding of the persons involved is needed. It seems to be a very heterogeneous one, which is the exact opposite of what you want when saying you want to avoid bias. You even speak of “geriatric patients” in line 293 – in which way were they patients???

2) Organization of the Manuscript: Line 24-30 is methods for me. Line 40-46 is methods to me as well. Line 53-80 again is methods. In Line 48, you say you give an overview on recent research. However, I am missing information on previous research using sensors in individuals’ homes (and I know there is such research out there; e.g. check work from Jeffrey Kaye).

3) you mention IMUs and GPS sensors in the methods section, but there are no results on this. Why is this even introduced? All your previous research mentioned (p 3) is concerning IMUs, but you do not analyse IMU data?!

4) Language/Style: the text at times is hard to follow as it is “jumpy” and volatile. For me, parts are really confusing. The thread is missing in chapters 1 and 2 – it goes back and forth. Partially its prose, even (e.g., line 92-93). I would also recommend to consult a native speaker. Examples: line 2: whether instead of “if”. Line 3: “mobility capacity and performance” instead of “cap. and perf. in mobility”.

5) One of my most problematic comments is regarding the display and interpretation of the results. Figure 3 is highly misleading, as it suggests a HUGE difference between groups. But in fact, a rise in SPPB scores of 0.16 points (line 246) is clinically absolutely irrelevant. 1.0 points would be something, but 0.16 is literally nothing. You should draw the figure with a scale of full 12 points, not 1.000 to 1.150 – that is eyewashing. Same with the TUG time. 0.1 seconds less is nothing. Even more problematic, you conclude that the exercise group has shown “greater improvement” (line 260-261). This is absolutely not true. Moreover, you would have to analyze this difference statistically, which you do not. Of course, with such a small sample, this would not make sense anyway.

6) The explanation of why there were only n=15 included in the analysis needs to come earlier, when the sample is explained. Otherwise it leaves the reader wonder. Also, you do not have a “cohort”, as your sample seems very heterogeneous to me. I think using the word “sample” is much more appropriate.

7) Sensor graphs: I find the description hard to follow, but that may be just me. Still, how were movements counted? Every X seconds? What if someone did standing balance exercises in a room vs. one walking around? Standing on one leg would probably not yield a hit on the motion sensors, still it would be a meaningful balance activity? It needs to be made clear what is counted at which intervals.

8) sensor graphs: I am in doubt that your motion sensors as applied here are able to capture changes as proclaimed in this work. Certain exercises as done in OTAGO would not be captured, visitor’s signals cannot be taken out. I wonder: why should anyone use this tool, given these flaws? I expect high costs as well, which further hamper its clinical prospects. I think it would make sense to turn the manuscript upside down and make a validation of the motion sensor system using the IMUs and GPS.

I do have many further minor points along the manuscript, but honestly, I think those major points would have to be addressed first and I cannot make the time to write down all minor points right now. Taken together, this manuscript in its present form is absolutely unclear, imprecise, and biased from its very design forward.

Reviewer 4 Report

The work is shown to be relevant to the research field, it is relevant and timely. Studies like this are needed to understand the performance of older adults in their daily lives through sensor graphs.
The paper conforms to the ethical and scientific standards required by the journal.
The research design is adequate, the data obtained present quality and statistical robustness and are presented in an appropriate manner. The complementary information presented is adequate for the purpose of the research. The design and presentation allow the reproduction of the research in other populations of interest.
The statistical tests provided in the research are consistent with the purpose of the research and fit the structure of the data, yielding sufficient significance.
The results, their interpretation and discussion of them, allow reaching valid and reliable conclusions.
The references are current and adequate in the field of study, not appreciating the lack of relevant studies.

Reviewer 5 Report

Overall, I think this manuscript adds significantly to the literature. Below are my comments

Introduction

When talking about the TUG and the SPPB it is worth mentioning that the TUG is a not a granular measure and based on the way it is scored one cannot tell whether there were differences/challenges in gait speed, balance, or muscle strength. If one has to determine which factor is changed/a challenge, one has to rely on qualitative assessment from clinicians observing the movement. Additionally, for the SPPB there must be a clinically relevant difference based on MCID for clinicians to use them to assess changes. 

Mentioning these limitations may help make a stronger case for why a computer science approach is needed.

Please complete sentence on line 89

Same with line 90

What do you mean by the sentence in lines 98 and 99?

Additionally, in your introduction it is worth mentioning that some studies have used RGB cameras to assess gait and balance but RGB cameras can be intrusive, while your approach is not. Some labs are also trying to use LIDAR cameras to detect these changes. Once again those have privacy concerns

Methodology

The methodology is well written, here are some comments

  1. You can combine Tables 3 and 4 and report whether there were statistically significant differences pre and post TUG and SPPB scores in those tables. Also in your tables please report the same information for the intervention and control groups and identify whether there were significant differences in age and frailty between participants at baseline. 
  2. How did you ensure that the participants performed the OTAGO once a week? Was it remotely administered?
  3. What were the dates that the data for this study was collected? 
  4. Was the 30-minute walk an outdoor walk or was it a treadmill walk? Was the walk at self-selected gait speed or was the gait speed monitored?
  5. Who conducted the assessments and tailored the exercise program? Was it a licensed physiotherapist? How did they determine whether exercise selection and intensity needed to be changed?
  6. How long did the questionnaires take? Were they administered before or after the SPBB and TUG? Boolani, et al (2020) reported that filling surveys led to a decrease in mental energy which led to clinically relevant declines in functional balance
  7. Where were the participants instructed to place the IMU sensors?
  8. Until the statistical analysis section the fact that you split the groups into control and intervention is very vague. Please make that clear at the beginning of the methodology section and identify what the control group did and how was it monitored that they were not as physically active as the OTAGO group

Results

Was there a statistically significant difference in improvement in SPPB scores in OTAGO vs control group?

Was the activity that's reported the activity measured by the sensors or by IMU?

Discussion

In your discussion you may want to touch on how TUG and SPPB limited in their measurements. You may also want to touch upon why both groups reported improvement in both. Was it improvements in functional capacity or was it learning effect? (especially since they performed these tests repeatedly)

Additionally, why did individuals in the control group also have improvements in TUG scores? 

You state that there was less activity in the flat. Could you hypothesize why that might be the case? 

Round 2

Reviewer 2 Report

1. Lines 130-140: the bullets of inclusion criteria and exclusion criteria can be the same style.
2. Line 212: It will be better to add a title for these tables.

Reviewer 3 Report

I do understand that some of my comments were very critical, especially given the fact that other reviewers recommended “accept” (which is a shame, to be honest; this has nothing to do with a proper peer review). Still, not all of my comments were addressed in an optimal way, and I still think that some aspects need to be changed in order to see this article fit for publication.

Comment 1: okay

Comment 2: won’t argue again here; still think it is not the best possible solution

Comment 3: okay, although I have rarely come across a manuscript describing things/methods not relevant to a manuscript because not used

Comment 4: okay

Comment 5: I disagree on the plot. The plot does give insight when plotted from 0 to 12 points: it would show that there is no difference, which also is an important information. I am not convinced by the authors’ response here. It is incorrect to change a scale to a degree that even a minimal insignificant difference appears meaningful. I suggest not to have a figure like this in a scientific manuscript.

Comment 6: We can twist and turn wording here. Although they might share some characteristics, combining community-dwellers with residential aged care residents and calling them a cohort to me seems incorrect.

Comment 7: If you are not measuring balance performance, you are missing one of the key aspects of performance in everyday life, especially when looking at OTAGO exercise which also focuses on balance. So how can you draw conclusions on performance in real life when missing this crucial part of performance? Also, balance is an important part of the SPPB (capacity measure) and –if changes in balance occurred – you would not be able to link balance performance and capacity. This at least would have to be discussed (very) critically!

Comment 8: I am aware that no causalities can be drawn from this study. Still, you did not answer my question on clinical prospects of this technology. I would expect a critical discussion of the limited practical value of the technology – at least I do not see one but if the authors do, this should be explained. In this form the reader wonders: what to do with it?!

Reviewer 5 Report

Thank you for addressing my comments.

A few more comments left to address

  1. Although there was a statistically significant improvement you should note that there was no clinically relevant improvement. In these sorts of clinical tests statistical significance may be meaningful to some but clinical relevance is more meaningful. With that being said, if there were any subjects who had a clinically meaningful improvement can you please report that and see how those individuals were different from the ones who reported no difference/decline in capacity
  2. I don't think you saw the correct Boolani, et al (2020) manuscript. The one cited below could be used as a citation because their subjects completed a series of surveys prior to performing the TUG and the BBT and reported a significant decline in BBT scores (including a clinically relevant decline in some participants). This may be one of the limitations of your study.
  3. Boolani, A., Ryan, J., Vo, T., Wong, B., Banerjee, N. K., Banerjee, S., ... & Martin, R. (2020). Do changes in mental energy and fatigue impact functional assessments associated with fall risks? An exploratory study using machine learning. Physical & Occupational Therapy in Geriatrics, 38(3), 283-301.
